# Data Budgeting for Machine Learning

## Abstract

Data is the fuel powering AI and creates tremendous value for many domains. However, collecting datasets for AI is a time-consuming, expensive, and complicated endeavor. For practitioners, data investment remains to be a leap of faith in practice. In this work, we study the data budgeting problem and formulate it as two sub-problems: predicting (1) what is the saturating performance if given enough data, and (2) how many data points are needed to reach near the saturating performance. Different from traditional dataset-independent methods like PowerLaw, we proposed a learning method to solve data budgeting problems. To support and systematically evaluate the learning-based method for data budgeting, we curate a large collection of 383 tabular ML datasets, along with their data vs performance curves. Our empirical evaluation shows that it is possible to perform data budgeting given a small pilot study dataset with as few as 50 data points.

## 1 Introduction

Collecting the appropriate training and evaluation data is often the biggest challenge in developing AI in practice. While the emerging data-centric AI movement has garnered tremendous interest and excitement in the research of the best practices of curating, cleaning, annotating, and evaluating datasets for AI, a critically important piece in the data-for-AI pipeline that is still under-explored is *data budgeting*: Currently, the investment of data remains to be a leap of faith: practitioners estimate their AI data budget mostly based on their experience. Systematic and principled approaches for estimating the number of data points needed for a given ML task are still lacking.

Throughout this work, we formulate **Data Budgeting** as two closely-related research problems: The first research problem, **Final Performance Prediction**, is to predict the saturating ML performance that we can achieve if given sufficient training data. This allows practitioners to gain insights on whether the ML tasks they are handling are *ML feasible* in their current forms. The second research problem, **Needed Amount of Data Prediction**, is to predict the minimum amounts of data to achieve nearly the saturating performance as if we are given sufficient data. This allows practitioners to know whether their ML tasks are *practically feasible* given their real-world data budget on data collection and data annotation.

Theoretical scientists have made great efforts to find the lower and upper boundaries for data budgeting problems on whatever common learning models or specific models. In application field, Power-law-related methods(fitting curves by $y = a + b \times x^c$, where $y$ represents the test result and $x$ represents the number of data points for training) are prevalent in solving such problems as introduced in Rosenfeld et al. (2019) and Johnson et al. (2018). Also, many works focus on problems requiring large training datasets such as Sun et al. (2017) and Kaplan et al. (2020) targeting deep learning models and language models, respectively. They tend to fit the accuracy curve and use the fitted curve to estimate the training effect with enough data. However, throughout our work, we investigate how we can learn from existing datasets and make data budgeting predictions for a brand new dataset. Our experiments show that there are common features regarding the datasets with similar data budgeting results. Another difference is that existing work on data scaling laws on pre-training foundation models on natural language and computer vision mostly looked at web-scale datasets. At the same time, we focus on tabular datasets, which often have smaller sizes and are more common in science and social science.

We have built a dataset of tabular datasets which contains 383 datasets for binary classification and multi-classification tasks. The sources of these datasets are from OpenML and Kaggle. We use the built dataset to verify that we can give the estimation of the two data budgeting problems for a new dataset through learning from other datasets. Then we propose a new method called Multiple Splitting to make up the lack of the test data in the research of pilot data and generate an array for each dataset. Later, we employ the machine learning method to explore the relationship between the basic features of the dataset and its real data budgeting.

Through the experiments, we find that learning from other datasets can help us make the prediction for the new datasets. Our empirical evaluation shows that it is possible to perform data budgeting given a small pilot study dataset with as few as 50 data points. What's more, we analyze the valuation of different features in solving these problems. Finally, we find some reasons that may lead to the wrong estimation of the final performance and some conditions where our methods don't perform very well.

## 2 Problem and Method

We formulate the data budgeting problem as: **(1) Predicting final performance**: what is the saturating ML performance that we can achieve if given sufficient training data, **(2) Predicting the needed amount of data**: the minimum amounts of data to achieve nearly the saturating performance as if we are given sufficient data. For a machine learning task $T$, we want to enable the prediction given only a tiny *pilot study dataset* (e.g., 50 labeled data, same distribution as whole dataset). The overview of the problem and the method is shown in Figure 1.

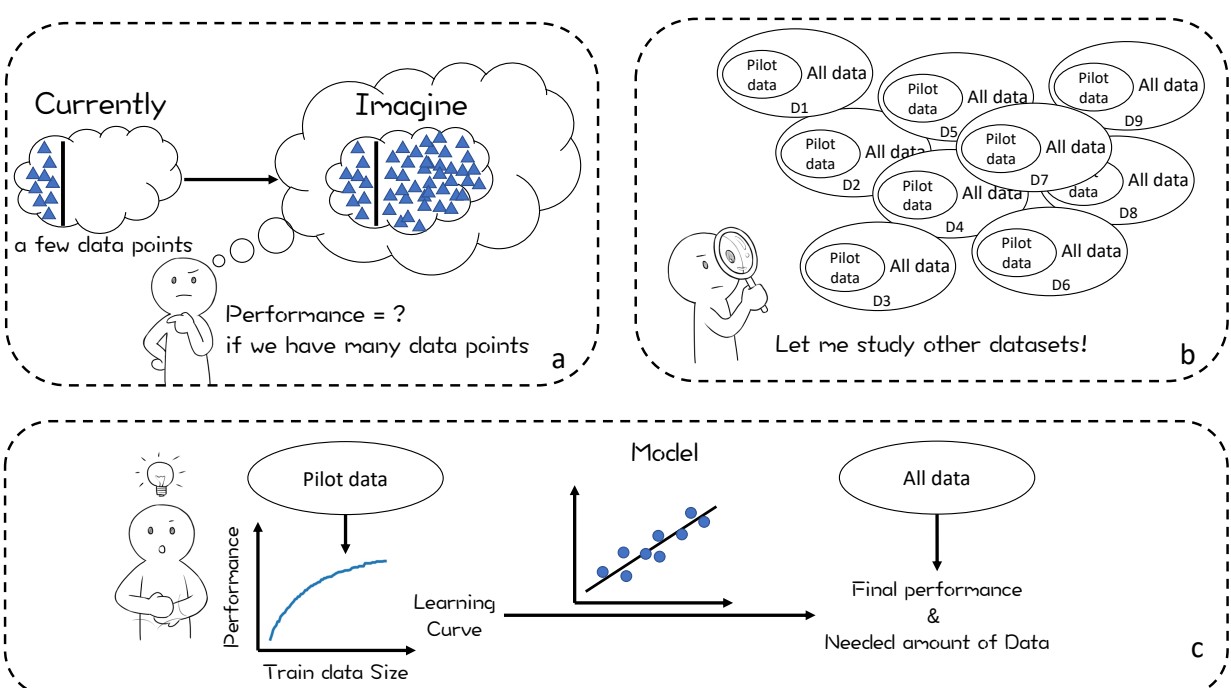

Figure 1: **Data Budgeting: Problem and Method.** (a) For a machine learning task, when we only have a few data points, we are curious about what will happen if we obtain more data points. (b) To solve such problems, we can refer to other existing big datasets. (c) We can abstract the problem as final performance and the needed amount of data prediction and quantify the pilot data by generating a learning curve related to the number of data points we use for train. Then we can learn two models to map the learning curves to final performance and needed amount of data separately. Therefore, we can give the prediction for tasks with only a few data points.

Formally, for one big tabular dataset $D$ with training set $D_{train}$ and test set $D_{test}$, we define the final performance of such dataset as the result of AutoML model trained on $D_{train}$ and tested on $D_{test}$. Though many AutoML Models with good performance exist, their difference is small compared to our estimation task. Here we utilize the combination of Autogluon library Erickson et al. (2020) and Auto-sklearn Feurer et al. (2015) by choosing the larger test result as the final test result. And the needed amount of data is the minimum amount of data that is sampled from $D_{train}$ where learning from such data can get nearly the same result as the final performance when tested on $D_{test}$.

Then we sample a few data points as a pilot dataset from each big dataset, and learn the relation between the pilot dataset and final performance / needed amount of data. In order to quantify the pilot data, we use an array $\vec{s}$ to represent it where $s_x$ represents the training performance when training with $x$ data points. Then we use the learning models such as linear models ( linear regression model or logistic regression model) and RandomForest models to find the mapping from $\vec{s}$ to the final performance / needed amount of data.

**The generation of $\vec{s}$** Since $\vec{s}$ should be generated with only pilot data, boosting the credibility of each $s_x$ is of great importance. **Single Splitting** method splits the pilot dataset once by sampling $x$ data points from the pilot data as train set and letting the remaining data points as test set. **Multiple Splitting** method repeats such operations for many times and calculates the average. Figure 2 suggests the number of repetitions be at least 500. We know that increasing the number of data for training will improve the training result. However, if we don't repeat sampling enough times, we will observe the fluctuation in the curve where training with more data will not contribute to better results, which means the result we obtain is not robust. And consider the efficiency, we don't find much difference between sampling for 500 and 1000 times. Therefore, repeating for 500 times is enough. Formally, for a pilot dataset $P$,

$$s_x = avg_{d \subseteq P, |d| = x} Result(Test\ on\ P \backslash d, Train\ on\ d)$$

We choose RandomForest Method as the training method($Train\ on\ d$). We explain why we choose RandomForest Method in the appendix: AutoML method doesn't work when data size is small; AutoML tends to choose RandomForest-related model as the best model.

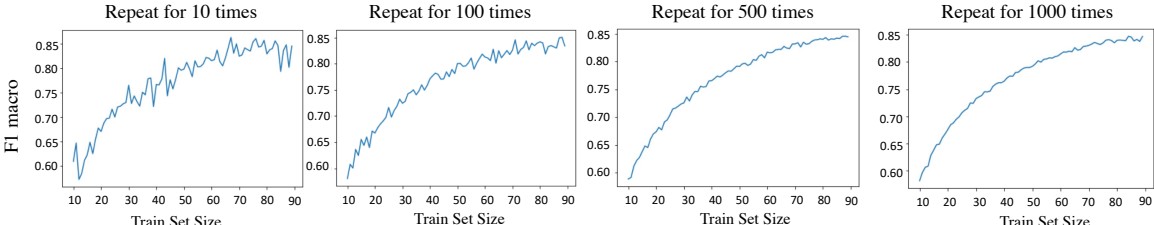

Figure 2: **The curves generated by repeating for different numbers of times.** We know that more data points for train lead to better test results. The fluctuation of the curve means that we haven't gotten robust results. We can see that increasing the number of repetitions will decrease it. And considering the trade-off between the curve quality and time spent, it is reasonable to repeat for 500 times.

**Final Performance Prediction** For a dataset $D$ with training set $D_{train}$ and test set $D_{test}$, we use Model $O$ (AutoML) to train it and calculate $F1_{macro}$ on $D_{test}$ where the metric $F1_{macro}$ is suitable for the real-world datasets which are commonly unbalanced. Finally, the final performance

$$O_D = F1_{macro}(D_{test})$$

Then we use linear regression model or RandomForest model to find the relation between $\vec{s}$ and $O_D$.

**Needed Amount of Data Prediction** We define $Need_D$ as the amounts of data we need to get nearly the same result as the final performance. That is to say, if we randomly choose a subdataset of $D_{train}$ whose

size is $Need_n$, we will get similar test results on $D_{test}$ when training with either this subdataset or $D_{train}$. Formally,

$$Need_D = argmin_n avg_{d \subseteq D, |d|=n} f(d) > 0.99 f(D_{train})$$

where $f(d)$ means the result tested on $D_{test}$ after training on $d$. Similar to the generation of $\vec{s}$, we use RandomForest model to train. Moreover, $f$ is not limited to the metric $F1_{macro}$. We extend it into a vector by adding the metrics Accuracy, F1_macro, Recall and Precision and redefine $\vec{a} > \vec{b}$ as each item of $\vec{a}$ is larger than the corresponding one of $\vec{b}$. By this way, we can give more stringent requirements for the subdatset.

## 3 Data Budgeting Benchmark

To support and systematically evaluate the learning-based method for data budgeting, we curate a large collection of 383 tabular ML datasets. The datasets are selected from OpenML(Vanschoren et al., 2013) and Kaggle(kag), which are two big repositories for machine learning and consist of a lot of tabular datasets. All the datasets' corresponding tasks are binary classification tasks or multi-classification tasks. We selected the datasets with more than 3000 pieces of data and the feature number of them is smaller than 50. The feature number should be smaller than expected pilot data size to avoid underfitting in the process of learning. Specifically, for the regression dataset, we change it to a binary classification task by dividing the labels into two bins from the median of the label.

**Datasets Summary**   In total, we have collected 383 datasets, covering diverse domains, including geo-science domains like volcano and finance domains like credit card application. These datasets constitute the new datasets $\mathcal{D}$. Among them, 330 datasets are from OpenML including 170 binary classification datasets, 95 multi classification datasets and 65 regression datasets. Moreover, 53 datasets are from Kaggle, including 42 binary classification datasets and 11 multi classification datasets.

**Datasets Preprocessing**   We first organized the datasets collected into the same format. Then for each dataset, we randomly chose 3000 data points, 500 of which were then selected as the test set $D_{test}$ to simulate the data distribution in reality. The remaining data points constituted the train set $D_{train}$. We used the test set $D_{test}$ to evaluate the power with one learned model. We kept the test set untouched when training the learned model. Then for each dataset, we generated a curve to depict the relationship between the power of the learned model and the number of data points to train such model. The curve's x-axis is the train set size while the y-axis is $F1_{macro}$. For $x = p$ on the curve, we randomly sampled $p$ data points from train set $D_{train}$, trained them with RandomForest Model, and finally tested the model on test set $D_{test}$. We repeated the process above multiple times and calculated the average of test results for each $p$ and each evaluation metric. We saved them for the experiments and possible future study.

## 4 Experiments and Analysis

### 4.1 Experiment Result

**Evaluation Method**   We split the datasets $\mathcal{D}$ for the verification of our method. Considering the similarity of some datasets' names and the possibility that their sources are the same, we divide the datasets into 100 clusters to ensure that similar datasets belong to one cluster. The details for clustering can be found in the Appendix. In the testing phase, we choose 80 clusters for training and 20 clusters for the test. Due to the limitation of current dataset $\mathcal{D}$ size, we employ bootstrapping by repeating choosing clusters for train and test for 40 times and calculating the average as the result.

In terms of metrics, for the final performance prediction problem, we calculate the $R^2$ value between the estimated value and the final performance $O_D$. For the needed amount of data prediction, we formulate it as a classification task where we predict the range of the needed amount of data for each datasets. We assign each dataset to one bin according to its needed amount of data. Specifically, we divide $0 - 2000$ into five bins: $[0 - 104, 105 - 227, 228 - 430, 430 - 805, 805 - 2000]$. And each bin contains nearly the same number

of datasets. We define $Acc_0$ as the accuracy of predicting the correct bin for the datasets. Moreover, we define $Acc_1$ as the ratio of the datasets where we mispredict them but the predicted bins are one bin near the correct bin.

**Baseline: Dataset-independent Method** Similar to the Powerlaw method introduced in Johnson et al. (2018). For one dataset, we fit $s_x$ into the function $f(x) = 1.0 - b * x^c$ where $x$ is the number of data points for train and $f(x)$ is learning performance with such data points. We use this fitted function $f(x)$ to predict the final performance and estimate the needed amount of data. Specifically, we use $f(2500)$ as final performance and find the minimal $x$ such that $f(x) > 0.99 \times f(2500)$ as the needed amount of data.

**Learning based Method** We use array $\vec{s}$ ($s_x = avg_{d \subseteq P, |d| = x} Result(Test\ on\ P \backslash d, Train\ on\ d)$) to predict the final performance and needed amount of data. We use two learning methods, linear regression and RandomForest, to find the relation between the $\vec{s}$ and final performance. For linear regression model , we fit the function $y = \vec{k}\vec{s} + b$ to find the $\vec{k}$ and $b$ where $y$ is $O_D$, as introduced in 2. And for RandomForest Model, we fit the tree to find the relation between $\vec{s}$ and final performance $y = O_D$.

Similarly, we can also find the relation between needed amount of data and $\vec{s}$. For logistic regression model, we learn the classification function $y = g(\vec{k}\vec{s} + b)$ where $y$ is the corresponding bin of needed amount of data introduced in 2. And for RandomForest model, we fit the RandomForest classifier to calculate the tree for $\vec{s}$ and $y$.

The comparison of Powerlaw method and learning method result is shown in Table 1 with different pilot data size. We find that learning method performs better than PowerLaw method, especially when pilot data size is small. Linear model outperforms RandomForest on final performance prediction while RandomForest has better performance on Needed Amount of Data Prediction.

When pilot data size is big, we tend to get more information about the datasets from the pilot data, thus the difference between Powerlaw and Learning is small. However, when pilot data size is small, though the information we can get is limited, we can refer to other datasets and learn some useful information for prediction. Therefore there is big difference between Powerlaw method and Learning method.

Though the performance of linear models and RandomForest is different, they all outperform PowerLaw. However, Linear models have better interpretability than RandomForest. We can choose to use linear model or RandomForest model according to our actual needs.

| Pilot Size | Final Performance Prediction ($R^2$) | | | Needed Amount of Data Prediction ($Acc_0$) | | |
|---|---|---|---|---|---|---|
| | Powerlaw | Learning(LR) | Learning(RF) | Powerlaw | Learning(LR) | Learning(RF) |
| 50 | 0.1714 | **0.5095** | **0.5188** | 0.1762 | **0.3673** | **0.4023** |
| 100 | 0.5500 | **0.6303** | 0.5687 | 0.1770 | **0.3988** | **0.4590** |
| 200 | 0.7465 | **0.7594** | 0.7385 | 0.2046 | **0.3961** | **0.4545** |

Table 1: Comparison of Power-law method and Learning method for different pilot data size. LR represents using Linear Regression/Logistic Regression model to fit whereas RF means using RandomForest/RandomForestClassifier to fit. We can see that learning method performs better than PowerLaw method, especially when pilot data size is small. Linear model outperforms RandomForest on final performance prediction while RandomForest has better performance on Needed Amount of Data Prediction.

### 4.2 Analysis

We will show some findings when doing the experiments. The results shown are under the setting that pilot data size is 100. However, the results under other pilot data sizes are similar to those under 100.

**Splitting the dataset more times leads to better results.** First we fix the pilot data size $m = 100$. We find the gap between the learning performance trained on 80 data points and the final performance. As shown in Figure 3(a), we compare four methods: (1) Single-Splitting: split the pilot data once, train on 80

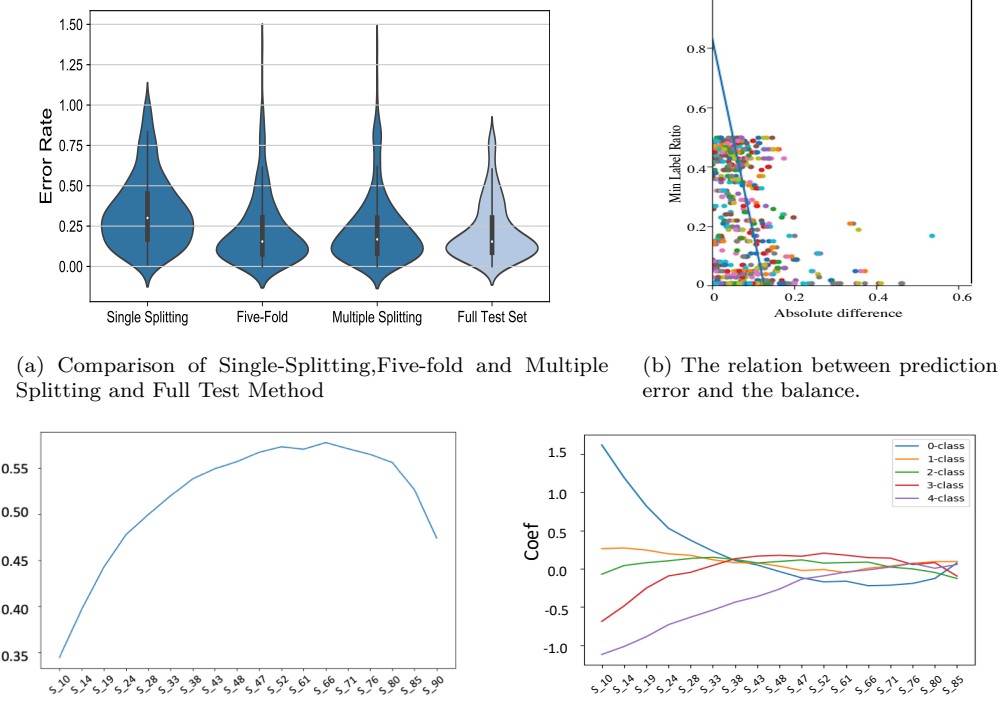

(a) Comparison of Single-Splitting,Five-fold and Multiple Splitting and Full Test Method

(b) The relation between prediction error and the balance.

(c) One point prediction

(d) Visualization of the linear model's coefficients.

Figure 3: **The Analysis of Data Budgeting Prediction. (a) Comparison of Single-Splitting, Five-fold and Multiple Splitting and Full Test Method** We define the error rate to be $abs(M_D/O_D - 1)$ where $M_D$ means the value under different method and $O_D$ is the final performance of dataset $D$. From the violin plot, we can see that methods except Single splitting tend to have smaller error rate. This indicates that adding test set size will decrease the gap between pilot data result and real final performance. **(b) The relation between the prediction error and the balance.** The x-axis represents the gap between the prediction value and real value ($abs(y_{predict} - y_{true})$) while the y-axis represents the ratio of the minority label in the pilot data (the proportion of the smallest category data in the overall data). Bigger y-axis value means more balance. The blue line fits these dots and shows that the proportional relation between the balance and error.**(c) One point prediction.**The picture shows the power of using linear model to map one $s_x$ to the model $O$ (AutoML). The y-axis indicates the $R^2$ between our prediction value and the $O$ (AutoML). The power of $s_x$ reaches the highest when $x$ is around 60 where the train set size is nearly the same as test set size. **(d)Visualization of the linear model's coefficients.**One point ($s\_x, y_{coef}$) represents the coefficient of $s\_x$ when using linear models to train the $\vec{s}$ array. For one certain dataset, if $s_x$ is large when $x$ is small, it tends to belong to class-0 (needed amount of data is small). On the other hand, is $s_x$ is small when $x$ is small and large when $x$ is big, it tends to belong to class-4(needed amount of data is big). Besides, we can see big difference for different classes in the early part of the curves. This indicates that the performance trained with minor data points is important.

data points and test on 20 data points; (2)Five-Fold: Split 100 data points into 5 folds and do the valuation for each fold; (3) Multiple Splitting, $s_{80}$ as is defined before; (4) Full Test Set: sample 80 data points from pilot data and test them on $D_{test}$(500 data points). We find that Full Test Set result is closer to the final performance where they share the same test set. This implies that the inaccuracy of predicting the final performance may come from the lack of test set. Without enough test data, it is hard to tell how it performs on the real data, let alone how close it is to the real result. Five-Fold method or Multiple Splitting method serves as the method to increase the test set size. Actually, they compensated for the lack of test data to some extent. Besides, Multiple Splitting Method can also generate an array and fully exploit the pilot data in the process of assigning data points to either train or test sets.

**The most informative part of the curve $\vec{s}$ is the center right.** Apart from using the whole $\vec{s}$ array for learning, we first use the simplest model: $y = ks_x + b$ for every chosen $x$ to learn the relation between $s_x$ and the final performance. From the observation in Figure3(c), we tend to easily find the relation between $s_x$ and the final performance when the training and test set size are nearly the same. On the other hand, when the difference of the train set size and the test set size is big, the power of prediction tends to decrease.

Inspired by such findings, we try to optimize the learning for the whole array. We print the coefficients for each $s_x$ and select some of the $s_x$ smartly based on the absolute value of their coefficients, i.e the importance of each $s_x$. We choose five $s_x(s_{10}, s_{14}, s_{19}, s_{52}, s_{61})$ with maximal absolute value for learning and get better $R^2 = 0.6634$. One possible explanation of the improvement generated by choosing $s_x$ with smaller $x$ is that training with minor points helps us know the primary power of the data, i.e., whether we can easily study the corresponding tasks. Besides, when balancing the train and test set size, we tend to have more accurate test results and know more accurate training effects. Another reason the result improves is that there may exist overfitting problems during the training.

**Imbalanced Datasets are harder to predict** We analyze which datasets are predicted incorrectly. As is shown in Figure 3(b), the imbalanced datasets tend to be mispredicted. The reason is that we will have few data for the minority label in the pilot data, thus lacking the cognition of these categories.

**Early part of the curve $\vec{s}$ is important for needed amount prediction.** To explain the power of our prediction, we visualize the linear model's coefficients in Figure 3(d). The 0-class, 1-class, ..., 4-class represent 5 bins $[0 - 104, 105 - 227, 228 - 430, 430 - 805, 805 - 2000]$ respectively to classify the datasets according to their needed amount of data. We find that if we can already achieve a high $F1_{macro}$ in the early part of the curve, then the needed amount of training data tends to be small. In contrast, if we observe significant performance improvement when adding more data points, the needed amount tends to be large. Besides, we can see a big difference for different classes in the early part of the curves. Such phenomenon, along with the findings in the property of $\vec{s}$, indicates that the performance trained with minor data points is important for data budgeting.

### 4.3 Generalization to Varying Pilot Study Dataset Size

The experiments in our previous experiments fix the size of the pilot data. However, in reality, the pilot data size is not fixed. We generate the pilot data for one dataset $D$ by first randomizing the pilot size $m$ and then sample $m$ data points from $D_{train}$. When solving the generalized problems, we treat pilot size as a new feature for training. Also, we define $s_{x\%} = s_{\lfloor x\% \times m \rfloor}$. Therefore for different datasets, we can generate the curves of the same length. For needed amount prediction, we redefine the label to be the needed number divided by the pilot size, i.e., the times of the size of the pilot data we need. Furthermore, we divide the datasets into 5 bins the same way as before. The result for the final performance prediction is 0.6997. For the needed amount of data prediction, the accuracy $Acc_0$ is 39.2% and $Acc_0 + Acc_1$ is 75%. The results reflect that our method can also tackle the situation when the pilot data size is not fixed.

## 5 Related Works

In the theoretical field, VC dimension (Wikipedia contributors, 2021) measures the power of a binary classifier by defining the VC dimension $m$ as the largest possible value $m$ where the classifier can separate $m$ arbitrary points. Also, VC dimension theorem gives the probabilistic upper bound on the model's test error related to the training set size. Meanwhile, sample complexity (Wikipedia contributors, 2022) gives the polynomial bound for the needed number of samples when the VC dimension of the model is finite. The required number of samples can be polynomial probabilistically bounded by test error. However, statistical learning theory proves that the upper bound of the difference between training error and generation error will be enlarged by the increase of capacity of the model(Goodfellow et al., 2016). That is to say, if the model is complex, it is less likely to give an estimation of the discrepancy between training and testing error. Moreover, though Kong & Valiant (2018) successfully estimates the accuracy when training with fix-distributed data on a linear

model, the actual data distribution seems like a black box to people. The judgment of the quality of data is also hard for people. Therefore, more application methods have been introduced.

In the application field, power law function has always been a good tool to fit the curves that depict the relationship between the number of samples and the performance, such as loss or error rate. Hashimoto (2021) gives evidence of the possibility of such a scaling model for the linear model and general M-estimators. And many works have been done to improve power-law model or make it more accurate for specific models. Hestness et al. (2017) shows the generalization error and model size growth has the scaling correlation with the size of training sets. Such scaling relationships give more insights for deep learning models on data budgeting problems. Rosenfeld et al. (2019) focuses on the neural network model by approximating the error rate with the number of samples and the width and depth of the neural network model. Such approximation derives from power-law. Besides, Johnson et al. (2018) introduces hyperparameters and proposes inverse square-root and biased power law in addition to traditional power law to give a more accurate estimation for NLP or ML model. What's more, Hashimoto (2021) refines the method of using log function to approximate the error function related to sample number, which gives a good method of predicting the simple model behavior.

For complex models, in the computer vision field, Sun et al. (2017) proves that increasing the number of samples can increase the performance of vision tasks by either directly feeding more data points into the model or training a better base model. It finds that the model performance increases logarithmically based on the training data size for tasks including image classification and object detection. Moreover, Kaplan et al. (2020) observes consistent scalings of language model log-likelihood loss with model size, dataset size, and the amount of compute used for training. It also points to the log relation between the dataset size and the performance in language models such as WebText2.

Besides log-linear related models, estimating Bayesian error rate(BER) of data distribution can help give the upper bound of the performance with enough data. The estimation of BER is difficult. Renggli et al. (2020) suggests using a simple 1-nearest-neighbor estimator on top of pre-trained embeddings to estimate it without introducing hyperparameters. And BER estimation not only helps predict the final performance but also serves as guarantees for website fingerprinting defences (Cherubin, 2017).

However, all the application methods introduced above are independent of other datasets. They seldom use the experiences of predicting data budgeting from other datasets while our work utilizes the information of those datasets.

## 6   Conclusion and Discussion

We discussed "data budgeting for Machine Learning", which consists of two questions: final performance prediction and needed amount of data prediction. Traditional methods are independent of other datasets. To address this limitation, we build a benchmark consisting of over 300 tabular datasets to make it possible to learn from other datasets. Our experiments verify that learning from other datasets has better effects than traditional dataset-independent methods. Also from the analysis, we find that the performance when training with minor data points is important for data budgeting prediction. The test set size and the balance of the labels also influence the prediction. Finally, data budgeting problems are complex because they are limited by the size of the pilot data and its incomplete display of real-world data. Such problems require further exploration by refining the model or introducing more datasets with the help of Active Learning.

**Broader Impact Statement**

Our work recognizes the ability to predict the data budgeting for one dataset from learning other irrelevant datasets. However, when evaluating the learning performance, we didn't carefully distinguish the datasets according to their label distribution, sources, etc. Actually, we can not guarantee that our method works well for all the datasets, especially those whose label distribution is severely uneven or whose features have too many dimensions. Moreover, the tabular datasets in our experiments are only subpopulations of all the tabular datasets. There may exist some datasets with specific properties being ignored by us. And sometimes the pilot data we get for a dataset doesn't have the same population as real data population.

Therefore people should keep eyes on the pilot data distribution when they use our method. In the future, more representative datasets should be intensively studied. Also, we can build and improve our model on more fine-grained objectives.

Admittedly, in the process of our experiments, we need to train the machine learning model many times to generate the training curve, which is vital for evaluation. However, generating the learning curves is also unavoided when using power-law methods. In the future, one possible strategy is to build a library containing the curves of those datasets to avoid double counting and help decrease the environment effect.

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

# A   Appendix

### A.1 Dataset Splitting

When noticing the name of the datasets we have collected, we have found that many of them have similar names. The source of data may be the same or they are describing the similar things. So if we want to verify the effects of datasets-dependent algorithm, we should not to arrange the similar datasets in both training and testing groups. For example, $House_8L$ and $House$ are two similar datasets describing the house price, we should put them either both in training set or in test set to assure that our training is robust.

To solve this problem, we first define the dataset similarity to be the SequenceMatcher value from difflib, which aims to find the longest contiguous matching subsequence that contains no "junk" elements. The sample heat map of the distance matrix can be shown in the Fig4. Then we do AgglomerativeClustering in the package Pedregosa et al. (2011) from this distance matrix and then cluster the datasets into 100 groups.

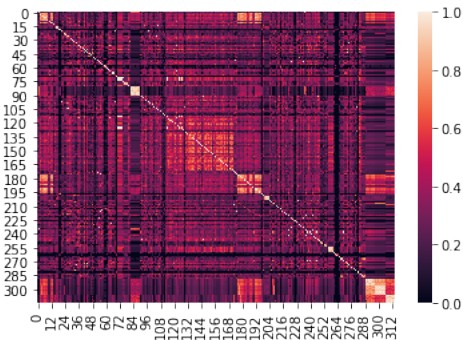

Figure 4: Heatmap

### A.2 AutoML and RandomForest

#### A.2.1 The selection of AutoML

There exist a lot of AutoML models. Here we show the difference between Autogluon and AutoSklearn on the datasets we build in Figure 5. We can see that the performance of Autogluon and AutoSklearn is nearly the same. Since our prediction is just to give a reference, the difference between different AutoML models can be ignored. In our work, we use the combination of Autogluon and AutoSklearn.

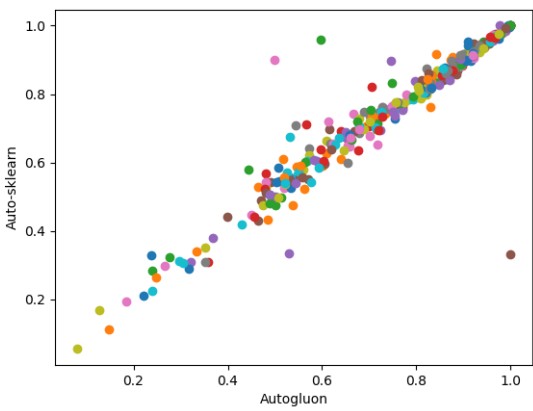

Figure 5: Comparision between Autogluon and AutoSklearn

### A.2.2 The reason we use RandomForest

In our work, we are required to generate curves. AutoML doesn't work well when data size is small. Therefore, to better know how AutoML works, we calculate the models selected in AutoML (Autogluon) for one dataset in Table 2.

Table 2: The AutoML's best model Statistics

| Method | Times to select as the best model |
| --- | --- |
| WeightedEnsemble_L2 | 323 |
| RandomForestGini | 14 |
| RandomForestEntr | 13 |
| LightGBMXT | 18 |
| LightGBM | 29 |
| CatBoost | 41 |
| ExtraTreesGini | 10 |
| ExtraTreesEntr | 10 |
| XGBoost | 25 |
| LightGBMLarge | 18 |
| KNeighborsUnif | 1 |
| KNeighborsDist | 1 |
| NeuralNetFastAI | 18 |
| NeuralNetMXNet | 12 |

We can see that Decision Tree based models(WeightedEnsemble_L2, RandomForestGini, RandomForestEntr',LightGBMXT, LightGBM, CatBoost, ExtraTreesGini, ExtraTreesEntr) dominate the best models. Besides, RandomForest Result is closed to AutoML result with $R^2 = 0.8551$. Since RandomForest is a simple model, it's easy to test, we choose RandomForest Model to approximate the training result($F1_{macros}$) for pilot data.

### A.3 Results for Other pilot size

To show the analysis in 4.2can be extended to the pilot data set size other than 100, we demonstrate the results for pilot data set size 50 and 200. And we can get similar results as the results of pilot data set size 100.

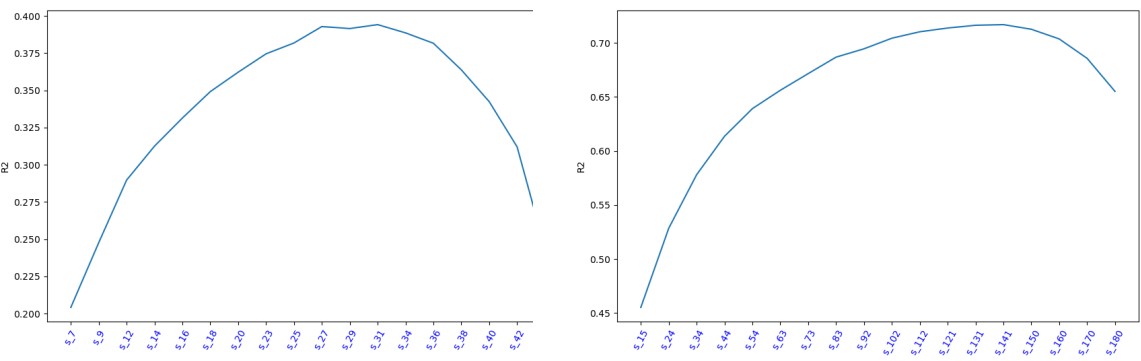

Figure 6: Pilot data set size $m = 50$          Figure 7: Pilot data set size $m = 200$

Figure 8: One point prediction. The picture shows use linear model to map one $s_x$ to the oracle power . The y-axis indicates the $R^2$ between our prediction value and the oracle.

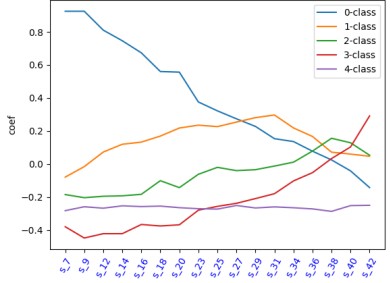

Figure 9: Pilot data set size $m = 50$

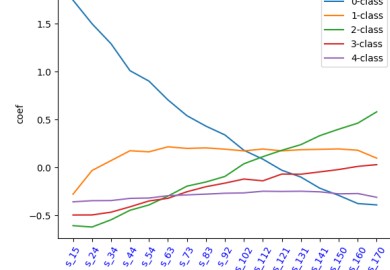

Figure 10: Pilot data set size $m = 200$

Figure 11: Visualization of the linear model's coefficients. One point $(s_x , y_{coef})$ represents the coefficient of $s_x$ when using linear models to train the s array.

