# OpenReview forum: "Data Budgeting for Machine Learning"
_TMLR — Rejected by TMLR_

### Review · Reviewer_jMxX · 2022-08-02

**Summary Of Contributions:**

The authors present an approach to a) measure the predictive performance that a machine learning (ML) model for tabular data sets can achieve and to b) estimate the number of data points needed to achieve this performance.

This motivation and the general idea are highly relevant and the approach taken by the authors is interesting. The structure of the paper is clear and the text is written such that the main ideas are easy to follow. There are some points, listed below, related to the experimental procedures and to the relevant related literature that could be clarified.

**Broader Impact Concerns:**

When selecting data given a limited labeling budget, often times the final predictive performance is not capturing all relevant aspects of the predictive performance. For instance, when there are protected groups of individuals affected by the decision of a ML model, discarding some parts of the data can lead to worse predictive performance for sub parts of the population. While it seems that accounting for fairness is not the main point of the proposed approach, it would still be necessary to raise awareness for these pitfalls when sub sampling training data.

**Requested Changes:**

* some more details on the ‘splitting method‘ would be helpful. It sounds like bootstrapping or plain k-fold cross-validation, but I’m not sure I’d be able to replicate this with the information provided in the paper.

* it would be helpful to see how the proposed method compares with other approaches that tackle the problem of limited labeling budgets, such as active learning, cover sets or transfer learning for HPO, see above. Maybe other sampling procedures would help to improve the proposed method or maybe other features, like vector space embeddings of data sets, could improve the estimate of the predictive performance?

* the authors could comment on how robust their estimate of the final predictive performance is, also in relation to the error of the proposed method, and how that impacts the proposed approach.

* it would be helpful to have a comparison of the linear regression method with some other non-linear method, maybe just the same AutoML approach, I think autogluon would detect that it’s a regression automatically so it would be a minor change of the implementation.

**Strengths And Weaknesses:**

A key factor for assessing the strengths and weaknesses of the proposed approach is probably the sampling of the data points. It does make a difference which data points are included when training an ML model, some are more important than others and will result in stronger gradient signals. This is the idea behind a number of research fields, including active learning, where the most useful data points given a certain labeling budget are identified, or research on cover sets, where the goal is to find a small number of data point that would result in the same predictive performance as when training on the entire data set.

Another practical consideration related to the sampling: not only the training data points but also the testing data needs to be chosen with care. If the test data is not representative enough, the predictive performance that is estimated, is likely to be wrong. Also, as the test set will be small, at least smaller than the training data set, it will be a biased sample in that when sampling a different test set the final predictive performance will be different. So it would be probably helpful to consider the distribution of test values when sampling multiple test sets, but in the paper it seems the authors only consider point estimates of small size test sets?

Another field of related work that is not as similar as the ones mentioned above, active learning and cover sets, is that of transfer learning for hyper parameter optimization. There are a number of methods that aim at estimating the performance of ML models (for a given hyperparameter candidate set and labeling budget) without running an actual training and evaluation iteration. Many of these approaches use a condensed version of a data set that allows to compute similarities between datasets, such as dataset2vec for instance. The models developed in this context allow for probing the classification performance in a similar manner as the proposed approach.

To summarize:

## Strengths

+ Simple and interesting approach to a relevant problem
+ Evaluation on a comprehensive suite of data sets

## Weaknesses

- Lack of clarity on the sampling
- Lack of clarity on the robustness of the estimate of the final predictive performance
- Relation to other relevant research fields could be explored a bit more / commented on

---

> ### Author Response · Authors · 2022-08-15
> **Response to Reviewer jMxX**
>
> Dear Reviewer jMxX,
>
> Thank you so much for the careful reading of our manuscript and the thoughtful feedback provided. We will do our best to address your concerns below and have revised the manuscript accordingly.
>
> ### Requested Change 1:
> Thanks for the comment. We think that our method is bootstrapping to solve the limit of the datasets currently available and make the experiments more persuasive. And we restate it in 4.1 part in the manuscript and some more details for clustering can be found in the appendix.
>
> ### Requested Change 2:
> Here we just focus on the tabular datasets which are full labeling. Indeed, limited labeling case is very important for data budgeting problem. But we want to focus on the full-label case in this work. Also, we have tried some features like dataset similarity to help our estimation, we haven't found improvement. Therefore we think the curve is somehow the most important for data budgeting problem in current setting.
>
> ### Requested Change 3:
> In the paper, we have demonstrated that our approach doesn't work well for imbalanced datasets. So the property of dataset itself will influence the result. And our evaluation is based on repeating many times. So we can use experiments to verify that learning (dataset-dependent method) can outperform traditional dataset-independent method.
>
> ### Requested Change 4:
> Thanks for your suggestion. We have thought about this question before and tried other methods. Before we thought that linear model has good interpretability and good performance on data budgeting problems. So we can use them to show that learning is a good way to solve data budgeting problem and improve the results.
> After carefully thinking about your suggestion, to make the work more complete, we have revised the manuscript by adding more results using RandomForest to show the possibility of using other models and the power of other methods. And different models have their own advantages in some cases, like we found that RandomForest have better results on needed amount of data prediction. And we think that people can choose different algorithms depending on their needs. For example, if they need good interpretability, they can choose to use linear model.
> And since the number of datasets we obtain now is limited, it's hard for us to use AutoML to find which is the most suitable method. Indeed, we have tried that 2/3-layer network doesn't work better than linear model. We think if we obtain more datasets in the future, we can use AutoML to find more characteristics of data budgeting problems.
>
> ### Broader Impact Concerns
> Thanks for your feedback. We add the discussion about sampling fairly in the Broader Impact Statement.

---

### Review · Reviewer_dLR8 · 2022-08-05

**Summary Of Contributions:**

The paper acknowledges the difficulty in collecting and maintaining datasets for AI/ML applications and categorizes the problems in two areas:

1. Final Performance prediction
2. Needed amount of data prediction

The work is based on a collection of 383 datasets collected from Kaggle and OpenML. Trained models to predict 1. and 2. perform better than data-independent methods.



**Broader Impact Concerns:**

Experiments done here are not broadly applicable; users have to train their own model repeatedly. Authors may point to other work that is transferrable, or discuss ways to address this.

**Requested Changes:**

Revise/resubmit after addressing the following:

- There are a few strong assumptions (As mentioned in the weaknesses section) that have to be explicitly justified. These are all strong assumptions and may impact the results of the paper.

-  Some explanations can be made clearer. For example, in the paragraph on page 4 starting with **Data Processing**, rather than what is presented as is, authors can easily start with "Specifically, for x = p on the curve, we sampled p...". Currently readers may be confused by "We first organized the datasets collected into the same format. Then we randomly chose 3000 data points for each dataset, 500 of which were then selected as the test set to simulate the data distribution in reality. The remaining data points constituted the train set."

- Another example of additional clarity required is from the following caption in Fig 2 - "Widely accepted is that more data points for train lead to better test results. Therefore, fluctuation should be avoided." What fluctuation should be avoided? Do you mean variance, or something else? Is this about some baseline or fitted curve? Why was a choice of 500 points made over 1000 or 2000 if fluctuation should be avoided?

- Comment on why power law is better than the ML model at higher pilot sizes. Also, is there a better baseline than power law?

- Other minor changes such as grammatical errors (like this - "The most informative part of the curve ⃗s is the late middle" , and formatting errors like "Figure3c", "and get better..", "unbalanced (or imbalanced?"), "unavoided"; page 6 looks like it has missing text. There are too many such examples to list here - please go through the entire paper and fix these

-  Lastly related works seems more fitting as part of the literature review up front

**Strengths And Weaknesses:**

## Strengths

- Enough datasets are considered to make general conclusions
- Experiments are done with sufficient rigor - for example, " In the testing phase, we choose 80 clusters for training and 20 clusters for the test and repeat the process for 40 times."


## Weaknesses

- Results are based on tabular data, and an AutoML library AutoGluon with the strong assumption that Autogluon will perform well for all problems. While Autogluon works well for many datasets - no free lunch; we can see from [1] that autosklearn, GCP, H2o, and others perform better on some datasets. Will be good to consider these results in some way

- Unclear why the choice was made to select only datasets with less than 50 features. This is a strong assumption and may affect final performance. This needs to be justified in the paper

- regression tasks (~65 datasets) are changed to binary classification tasks - why not just have a relevant metric like RMSE or other for regression tasks. I am sure you can change $F1_{macro}$ to $loss$ where $loss$ is selected based on the problem type. Based on your paper it can also be a vector

- For the needed amount of data prediction, "we formulate it as a classification task by splitting the prediction targets into 5 equal-sized bins We use the accuracy $Acc0$ as the ratio of making the correct prediction of the bins the datasets belong to." - This and the use of $R^2$ needs to be justified

-



[1] https://arxiv.org/abs/2003.06505

---

> ### Author Response · Authors · 2022-08-15
> **Response to Reviewer dLR8**
>
> Dear Reviewer dLR8,
>
> Thank you so much for the careful reading of our manuscript and the thoughtful feedback provided. We will do our best to address your concerns below and have revised the manuscript accordingly.
>
> ### Requested Change 1
> Thanks for your suggestions.
>
> For weakness 1, we have slightly modified how we select AutoML model. We demonstrate in the appendix that the difference between different automl models (autogluon and auto-sklearn) is small compared to our prediction tasks. However, we modified our automl model as the combination of autogluon and auto-sklearn. Automl model here is just a reference to depict how well we can get in real world with enough data.
>
> For weakness 2, we have explained why we choose the dataset with less than 50 features in our paper in the section 3. Actually, we chose the dataset with less than 50 features to avoid underfitting in the training.
>
> For weakness 3, we think that there's a big gap between RMSE for regression task and f1_macro for classification task. So if we train our model with classification tasks, it will not work for regression tasks. Here we just want to increase the number of datasets to make our model more stronger. Therefore we don't think separate them is useful.
>
> For weakness 4, we reclarify it in 4.1 'Evaluation Method'. Thanks for the suggestion. We try to make it more clear to the readers.
>
> ### Requested Change 2
> Thanks for the suggestion. We have revised  part 4.1 to make it more clear to the readers.
>
> ### Requested Change 3
> Thanks for the suggestion. We have revised this part (Figure 2 related). We want to show that the fluctuation of the curve means that we haven’t gotten robust result. Therefore we need to repeat it for more times. And when we generate these curves, we should consider about time consuming and efficiency. Therefore we choose a suitable value, 500. Details can be found in the manuscript. Thanks.
>
> ### Requested Change 4
> We add the discussion in the 4.1 Learning based methods. And according to other reviewers' suggestions, we have revised this part by comparing it with more algorithms. And some complex powerlaw-related method is for big datasets of specific models (we mention it in the related work). We don't think it is easy to compare our methods with them.
>
> ### Requested Change 5
> Thanks for the suggestion. We have revised them and will continue to check our manuscript.
>
> ### Requested Change 6
> Thanks for the suggestion. To make our manuscript more readable, we received your suggestion and summarized some of the related works in the introduction part. Hope this change will make our manuscript more logical.

---

### Review · Reviewer_4oZU · 2022-08-07

**Summary Of Contributions:**

Data is the heart of any machine learning or AI use case. Most companies are not able to implement the AI goals as they think that they don't have enough data or maybe they are not aware of how much data is actually required to produce an impactful result. The research presented in this paper is ground-breaking and could address one of the biggest concerns of many startups. The research presents two novel approaches on how to define the "stable performance" given enough data for a problem and how much data is required to reach that stage. The research was conducted using 383 ML tabular datasets which were procured from Open ML and Kaggle. They also presented the challenges faced when handling unbalanced datasets over balanced datasets. The training technique used by the author is using AutoML. The paper clearly reflects on the model used in AutoML to facilitate the research. the results are quite intriguing and highlight some of the important aspects such as the spitting of the dataset multiple times leads to better results.

**Broader Impact Concerns:**

No ethical concerns

**Requested Changes:**

I have a few questions/concerns, if you can provide me some insights I can decide if they should be included to strengthen your research:

1. Did you find any related work or research trying to solve this data budgeting issue with an ML approach? I see you already mentioned the simple heuristics method but I am curious if anyone tried to find a solution using ML but might have faced certain challenges that you are trying to address in this paper as well.

2. Is there a reason to choose Auto ML? And not leverage some other framework?



**Strengths And Weaknesses:**

Strengths
1. The visual(figure 1)  to explain the problem is quite useful to understand the goals of the author
2. The comparison with Power Law Method states the success of the research and is a good reference point for the reader
3. Great work in highlighting a generalized approach to handling different pilot study sizes as that is how the real-life data would work

Weaknesses
No weaknesses as such

---

> ### Author Response · Authors · 2022-08-15
> **Response to Reviewer 4oZU**
>
> Dear Reviewer 4oZU
>
> Thank you so much for the careful reading of our manuscript and the thoughtful feedback provided. We will do our best to address your concerns below and have revised the manuscript accordingly.
>
> ### Requested Change 1
> We haven't found some ML-related methods. If there exists some related works, we are happy to refer to and compare with. And since we haven't found some related datasets, the possibility of related works existing is small.
>
> ### Requested Change 2
> We just use AutoML as a reference (oracle) for our tasks. We want to know how good we can predict the dataset in real world. We add the comparison of Autogluon and auto-sklearn in the appendix and discuss that we think the difference can have limited influence on our work.
> But we also finetune our AutoML model by combining Autogluon and Autosklearn and revised the results in the manuscript.
> Notice that when dealing with pilot data (data size is 50,100,200), we don't use AutoML model for the data size is small (hard to separate validation set). We use RandomForest and give the explanation in the appendix.

---

### Review · Reviewer_EJaN · 2022-08-14

**Summary Of Contributions:**

The paper proposes a method for studying the data budgeting problem in ML. Data budgeting is defined as a two-fold problem 1) predict the saturating generalization performance of a ML model given sufficient training data, and 2) predict the amount of needed training data to achieve the saturating performance point.

The authors use a very large number (383) of tabular data sets from OpenML and Kaggle to study the efficiency of their proposed method. Tabular datasets are widely used, thus studying the data budgeting problem for this type of data is important.

The authors propose to solve the data budgeting task for new data sets by leveraging data budgeting information from old data sets. Some insights of the paper, although which could be considered well-known, are useful: 1) to analyze the performance of a model when the data budget is small, it is important to employ methods such as k-fold (nested cross) validation, and 2) data budgeting is an even more difficult problem for datasets with high class imbalance. Another interesting insight is that power-law approaches may be more appropriate for cases with larger amounts of data, while a learning approach, as the one proposed in the current paper, may be more suitable when only very small amount of data (less than 200) is available.


**Broader Impact Concerns:**

Making the 383 data sets available together with their performance curves would be a good contribution to the community. This would also help with the reproducibility of the results. However, it is not clear how could one reuse these results, because the ML model could be different from the one used in the paper. In the case of the latter, one would need to build the performance curves from scratch.

I do not see ethical concerns.

**Requested Changes:**

- Please address my questions and comments from the previous review section.

- The terminology of the paper could be clearly defined from the beginning: pilot data, final performance (I find test or generalization performance better than final performance).

- 'k' is defined but never used in the experimental results section. It could be used for instance when the authors refer to the model coefficients.

- I encourage the authors to review the paper and fix the grammatical / punctuation errors. Also, the quality of the images could be improved (especially Fig. 3b). Finally, it should be explained how the y axis values are computed in Fig. 3 (for a single data set or averaged across multiple data sets).

- What does "late middle" mean in Section 4? This is unfortunately not clear.



**Strengths And Weaknesses:**

The paper addresses a very relevant topic for the ML community. Also the study is quite thorough, using many open datasets and providing many experimental results. The method is also simple and looks promising. I would have however several questions and comments that I think would help clarify different aspects of the paper and which would improve the quality of the paper if properly addressed in the manuscript.

- The method (Section 2): the main novelty of the paper claimed by the authors is that one could leverage the data budgeting analysis from old datasets to perform data budgeting for new datasets. It is not clear however from the description of the method how the method does it. I really like and appreciate Fig.1, but Fig 1c could better explain 1) what are the inputs and outputs of the method, and 2) how to transfer the knowledge from some datasets to newer datasets.

- The sampling: how is the sampling actually done? Are the samples (train/pilot, and test) representative of the original data distribution?

- It would be good if the authors would provide more details about: 1) why only datasets with less than 50 features have been chosen, 2) why linear regression has been chosen to learn the performance from the data size (R2 of 0.6 does not seem to indicate a linear relationship, maybe another model could perform better than the one used in the paper), and 3) is the same model type (Random Forest) tuned for all datasets? It is known that for tabular datasets gradient boosting models provide typically better results than random forest models.

- Auto-Sklearn, H2O, AutoAI are just a few examples of popular AutoML tools. Could the authors provide more details about their choice of the AutoML tool?

- The regression datasets have been converted to classification datasets. This requires justification. Is there any limitation in the proposed method regarding the task type (classification/regression/ranking etc)?

- Table 1: what happens when the data size increases above 200? Is the power-law approach better for the prediction of the needed amount of data too? It would be good if the authors could explain (or give some insights regarding) why power-law approaches are better when larger data budgets are available.

---

> ### Author Response · Authors · 2022-08-15
> **Response to Reviewer EjaN**
>
> Dear Reviewer EjaN,
>
> Thank you so much for the careful reading of our manuscript and the thoughtful feedback provided. We will do our best to address your concerns below and have revised the manuscript accordingly.
>
> ### Requested Change 1
> Weakness 1:
> According to your suggestion, we have added the description of inputs and outputs in the description for Fig 1(c) and in the text we use ‘mapping’ to indicate the inputs and outputs for the learning.
>
> Weakness 2:
> The sampling here is all random and we want to imitate real world data distribution. We clarify this in section 3 ‘Datasets Preprocessing’ part.
>
> Weakness 3:
> Thanks for the suggestion. We have made such modifications:
> (1) explain why we choose less than 50 features in the manuscript. The main concern is to avoid underfitting since pilot data size is small.
> (2)/(3) We have added the  method like Randomforset in the experiments and made comparisons between Randomforest and Linear model. The reason why we choose linear model are that linear model has good interpretability. And we want to show that data budgeting problem can be solved by learning.
>
> Weakness 4:
> We just use AutoML as a reference (oracle) for our tasks. We want to know how well we can predict the dataset in real world. We add the comparison of Autogluon and auto-sklearn in the appendix and discuss that we think the difference can have limited influence on our work.
> But we also finetune our AutoML model by combining Autogluon and Autosklearn and revised the results in the manuscript.
>
> Weakness 5:
> In our work, we only deal with classification task and transfer regression task to classification tasks. The reason is that we should keep the evaluation the same (Like f1_macro). However, if we have enough regression tasks, we can also make  the model with all regression tasks.
>
> Weakness 6:
> Thanks for the suggestion. We have added the explanation for the results with different pilot size. When pilot data size is big, we can get more information from the curves without learning other datasets. So powerlaw will have good performance.
>
> ### Requested Change 2:
> Thanks for the suggestion. We added more explanation for pilot data and final performance in the text.
>
> ### Requested Change 3:
> Thanks for the suggestion. We have revised them and will continue to check our manuscript.
>
> ### Requested Change 4:
> We have change this word to ‘center right’. This means that the important part is on 60%-80% percent position of the curve. I am not sure this change will answer your concerns.

---

### Review · Reviewer_u5La · 2022-08-15

**Summary Of Contributions:**

This paper studies the “scaling law” for AutoML models on open-source tabular datasets, and focuses on predicting both the performance with a certain number of available data points in a dataset, and predicting the maximum performance and the number of data points needed to achieve such performance - the data budgeting problem. Such laws may be informative to machine learning and data science practitioners to determine the amount of data needed and predict final performance of the model.

**Requested Changes:**

Answers and associated changes to the following questions should be critical for the paper to be convincing and valuable to the machine learning community:

1. Why study the data budgeting using AutoML models (according to the top of Page 3), not standard and simpler ML models like logistic regression, random forest, simple feedforward networks, etc.? This seems to be an extra and kind of redundant layer on top of the problem this paper studies (because of implicit biases of such AutoML methods, for example), and also limits the scope of this paper. The information obtained by studying data budgeting on standard ML models can also be applied to that on AutoML methods (since those methods select among standard ML models), but not the opposite.
2. What’s “one point prediction” in Figure 3(c)? There does not seem to be an introduction or definition. Also, what are “0-class” to “4-class” in Figure 3(d)?
3. Many figures have lines that do not have error bars to show information like confidence intervals, making the figures taking a lot of space without conveying enough information. I’d suggest the authors add such additional information.
4. What also concerns me is that the R2 numbers in Table 1 and beyond are kind of small, especially for those <0.5. An R2 this small does not seem to indicate a strong scientific relationship (I appreciate the authors’ candidness of course). This relates to my concern in Figure 3(b), in which the fitted line seems to come out of (x, y) points with a really weak relationship in between x and y values.
5. The authors should list the 383 datasets used in this project for reference. I’d expect there to be at least some datasets that are not too small (say less than 2,000 data points, for example).
6. The “Datasets Preprocessing” paragraph in Section 3 needs format checking, proofreading, and polishing. It’s hard for me to understand how the dataset is splitted.

**Strengths And Weaknesses:**

Strength: The problem this paper studies is interesting and important to the ML community.

Weakness: The design of experiments and presentation of results need more work for the results to be convincing. See Requested Changes below.

---

> ### Author Response · Authors · 2022-08-19
> **Response to Reviewer u5La**
>
> Dear Reviewer u5a,
>
> Thank you so much for the careful reading of our manuscript and the thoughtful feedback provided. We will do our best to address your concerns below and have revised the manuscript accordingly.
>
> ### Requested Change 1
> Thanks for the suggestion. Here we focus on two questions: final performance prediction and needed amount of data prediction. For Needed amount of data prediction, we focus on simple model like randomforest for. However, for final performance prediction, we want to give more accurate results in the real world. Automl doesn’t suit the datasets with the small number of data points for it’s hard to separate the train and validation set. But we can also use randomforest method to generate curves to estimate them. This is what we want to convey.
> And we have tried to use randomforest results as benchmark to replace AutoML. But we don’t think it’s necessary to show in the text. Maybe we can discuss this problem.
>
> ### Requested Change 2
> For Figure 3(c) one point prediction, we explain it in the text ‘Apart from using the whole $\vec{s}$ array for learning, we first use the simplest model: $y = ks_x + b$ for every chosen $x$ to learn the relation between $s_x$ and the final performance.’  in the ‘The most informative part of the curve $\vec{s}$ is the center right.' part. And for 0-class and 4-class problems, it represents the 5 bins that we classify the needed amount of data into.
> Thanks for your suggestion, we added more explanation in the text and caption.
>
> ### Requested Change 3
> We show the figures to do some analysis. Indeed we put some pictures for pilot size = 100 in the text and put other pictures for the different pilot size in the appendix. We want to show some analysis for the results and use these figures to explain. Our main results are in the table.
>
> ### Requested Change 4
> Figure 3(b) is not used to represent the result. We just  indicate that which dataset doesn’t perform well in our case, so we show the correlation between error rate ant imbalance. And we found that imbalanced datasets tend to has worse performance.
>
> ### Requested Change 5
> Here we choose the datasets with more than 3000 data points to make our work more persuasive. The referenced datasets and code can be found here https://anonymous.4open.science/r/Dataset-Estimation-5021
> And we will continue to complete it.
> Thanks.
>
> ### Requested Change 6
> Thanks for your suggestion. We have reviewed it to make it more readable.

---

### Decision · Action_Editors · 2022-09-02

**Recommendation:** Reject

**Comment:**

The submission tackles a very important problem for ML community: for a given task, one has to efficiently estimate the upper bound on the possible performance and the amount of labeled datapoints (data budget) needed to achieve this upper bound. Basically, the authors aim to demonstrate that it is possible to obtain a "cross-task" model that can be "trained" on a diverse set of tasks (for which the upper bound and data budget can be obtained in a brute-force way via extensive usage of the state-of-the-art AutoML frameworks) and then can generalise on the unseen tasks, where only a few datapoints are available.

An important advantage of the submission is that it tackles an extremely important research question since the positive results on this topic would be appreciated by many practitioners. However, after reading the paper and the reviews I cannot recommend to accept the paper in its current form. The main concern is about very low values of R2 and accuracy in the Table 1, which mean that the authors's model does exhibit some generalisation but it is hardly applicable due to its low predictive ability. Therefore, the paper does not provide an affirmative answer if the data budgeting problem can be solved to the degree of the practical use. The reviewers also had multiple concerns on the lack of clarity and explanation for several design choices. AE shares these concerns as well but considers them being less important compared to the discouraging quantitative results in Table 1.